

# Simultaneous and co-located wind measurements in the middle atmosphere by lidar and rocket-borne techniques

Franz-Josef Lübken[1], Gerd Baumgarten[1], Jens Hildebrand[1], and Francis J. Schmidlin[2]

[1]Leibniz-Institute of Atmospheric Physics, Schloss-Str. 6, Kühlungsborn (URL: www.iap-kborn.de), Germany
[2](Emeritus) NASA, Goddard Space Flight Centre, Wallops Island, Virginia, USA

*Correspondence to:* Franz-Josef Lübken (luebken@iap-kborn.de)

**Abstract.** We present the first comparison of a new lidar technique to measure winds in the middle atmosphere, called DoRIS (Doppler Rayleigh Iodine Spectrometer), with rocket-borne insitu observations which rely on measuring the horizontal drift of a target ('starute') by a tracking radar. The launches took place from the Andøya Space Center (ASC), very close to the ALOMAR observatory (Arctic Lidar Observatory for Middle Atmosphere

Research) at 69°N. DoRIS is part of a steerable twin lidar system installed at ALOMAR. The observations were made simultaneously and with a horizontal distance between the two lidar beams and the starute trajectories of typically 0-40 km only. DoRIS measured winds from 14 March 2015, 17:00 UTC to 15 March 2015, 11:30 UTC. A total of 8 starute flights were launched successfully from 14 March, 19:00 UTC to 15 March, 00:19 UTC. In general there is excellent agreement between DoRIS and the insitu measurements considering the combined range

of uncertainties. This concerns not only the general height structures of zonal and meridional winds and their temporal developments, but also some wavy structures. Considering the comparison between all starute flights and all DoRIS observations in a time period of ±20 min around each individual starute flight, we arrive at mean differences of typically ±5–10 m/s for both wind components. Part of the remaining differences are most likely due to the detection of different wave fronts of gravity waves. There is no systematic difference between DoRIS

and the insitu observations above 30 km. Below ∼30 km winds from DoRIS are systematically too large by up to 10-20 m/s which can be explained by the presence of aerosols. This is proven by deriving the backscatter ratios at two different wavelengths. These ratios are larger than unity, which is an indication for the presence of aerosols.

## 1 Introduction

Wind measurements in the stratosphere and mesosphere are crucial for our understanding of basic physical pro-

cesses in the middle atmosphere. This concerns, for example, the general circulation of the atmosphere and its impact by gravity waves, some of which are filtered by the background wind when propagating from their tropospheric source to the main breaking region in the middle atmosphere. Winds also transport chemically active trace constituents over large distances, as is evident, for example, by the global distribution of stratospheric ozone and water vapor. Therefore, winds can indirectly impact the energy and momentum balance of the atmosphere, and are



themselves largely determined by the general thermal structure of the atmosphere (thermal wind relation). In this paper we concentrate on the upper stratosphere and lower mesosphere, i.e., roughly from 20 to 65 km.

Several techniques to measure winds are applied in the middle atmosphere. However, they are limited in terms of altitude coverage and/or spatial and temporal sampling. Radar techniques rely on backscattered signals which, in the ionosphere, require the existence of free electrons. This technique therefore works only for altitudes above ∼70–80 km (see, for example, Hocking, 2011, and references therein). More recently, microwave instruments are used to measure winds in the stratosphere and mesosphere, however, with a comparatively poor altitude and time resolution (Rüfenacht et al., 2012). Satellite borne experiments for wind observations measure Doppler shifts of atomic and molecular emission lines which is limited to altitudes above approximately 80 km (see, e.g., Shepherd et al., 1993). The thermal wind relation is commonly used to deduce winds from a measured temperature field. Apart from the assumption of geostrophic balance, this method is rather limited in terms of altitude resolution and temporal coverage.

Perhaps the most reliable method to measure winds in the mesosphere and lower thermosphere (MLT) is based on the drift of a target which is transported into the MLT region by a rocket and then followed by a tracking radar. Various targets have been used in the past, such as falling spheres, chaff clouds, and so called starutes (STAble Retardation parachUTE) (Schmidlin, 1985; Widdel, 1990). For this paper we use starutes which are part of a datasonde launched by Super-Loki rocket motors. The obvious disadvantage of rocket-borne techniques is that they can only be employed sporadically and therefore cover a limited time period.

We have developed a new lidar technique to measure winds in the stratosphere and mesosphere which is based on quantifying the Doppler shift of the Doppler broadened backscattered Rayleigh signal. The technique is called DoRIS (Doppler Rayleigh Iodine Spectrometer) and is described in detail in Baumgarten (2010). DoRIS is part of the Rayleigh/Mie/Raman (RMR) lidar of the ALOMAR (Arctic Lidar Observatory for Middle Atmosphere Research) observatory in Northern Norway (69°N) which actually consists of a double lidar system and allows to measure temperatures and winds in two directions simultaneously (von Zahn et al., 2000). A first comparison of winds measured by DoRIS and by a sodium lidar also being located at ALOMAR showed good agreement in the limited height region of overlapping measurements around 80-85km (Hildebrand et al., 2012). ALOMAR is located very close to the Andøya Space Center (ASC) which offers the unique opportunity to compare winds from DoRIS with in situ measurements applying rocket-borne techniques.

The purpose of this paper is to report the first comparison of winds measured by DoRIS with in situ observations by datasondes launched from ASC on 14/15 March 2015 in the frame of the WADIS-2 (WAve propagation and DISsipation in the middle atmosphere) campaign.



## 2 Instrumental techniques

DoRIS measures the Doppler shift of the Rayleigh signal as part of the RMR lidar at ALOMAR. The laser wavelength of approximately 532 nm is tuned to an iodine absorption line. Backscattered photons are transmitted through an iodine cell. The transmissivity of the iodine cell depends on the Doppler shift of the backscattered

signal relative to the iodine absorption line. The amount of photons passing the iodine cell is therefore a measure of the Doppler shift, and thereby of the wind velocity along the line-of-sight of the lidar. The signal also depends on the Doppler width and thereby on atmospheric temperatures (see Baumgarten, 2010, for more details). It is important to note that we use temperatures as measured by the same lidar. The two telescopes of the ALOMAR twin lidar system were pointed off-zenith to achieve an optimal overlap with the datasonde trajectory. The so called

'North-West telescope' (NWT) was pointed to the north with a zenith angle of 30 degrees, and the 'South-East telescope' (SET) was pointed to the east with a zenith angle of 20 degrees. In this paper we consider wind profiles from DoRIS averaged over 15 minutes, sequentially shifted by 5 minutes, and sampled within an altitude bin of 2 km, sequentially shifted by 150 m. Wind errors for DoRIS are mainly due to the statistical noise of the signal. They are about 4 m/s, 3.5 m/s, 9 m/s and 18 m/s at 30, 40, 60 and 70 km altitude, respectively. We note that the

presence of aerosols distorts the wind measurements of DoRIS since they cause an extra signal at the center of the Doppler broadened line (Baumgarten, 2010). This will be discussed in more detail in section 5.

Wind measurements using the small Super-Loki rocket depend on the ejected payload containing a datasonde and starute. The starute is ram-air inflated after it is ejected into the mesosphere, often above 70 km altitude. The system has a total weight of 0.655 kg (starute: 0.155 kg, payload: 0.5 kg), a nearly quadratic-shaped cross section

(width 2.13 m) and a cross section area of 4.26 m$^2$. The starute contains a metalized burble fence which stays inflated by the air forced through the starute and aids in providing stable performance (no pendulation as found with typical parachutes). The trajectory of the starute is derived from a tracking radar following the target. In Fig. 1 a typical starute trajectory is shown together with the two lidar beams of DoRIS. A more quantitative presentation is given in Fig. 2. As can be seen from these Figures, typical distances between the lidar beams and the starute

trajectory are on the order of 10–30 km in the altitude range 30–60 km. The starute also carries a small thermistor to measure temperatures below approximately 60-70 km. In our case, these measurements were not achievable due to long-term technical degradation of the thermistors and the electronics.

Within the WADIS-2 campaign period we launched a total of 13 datasondes from the Andøya Space Center. A list of launches is shown in Table 1. In this paper we concentrate on the eight successful flights (with some

reservations) performed in the night 14/15 March 2015. In this period DoRIS was in operation from 17:00 (14 March) to 11:30 UTC (15 March), both with day-time and night-time measurements. In this paper we concentrate on night-time observations between 18:30 and 03:30 UTC.

During WADIS-2 several radiosondes were launched which also provide wind information, however only up to approximately 30 km. A list of radiosonde launches relevant for this paper is provided in Table 2. We noted a



systematic and regular oscillatory fluctuation in all winds measured by radiosondes. We removed this oscillatory signal by smoothing. The radiosondes typically require 1–1.5 hours before they reach their highest altitudes. In this time they drift horizontally by up to 100–150 kilometers.

## 3 Uncertainty analysis

Following the trajectory of a lightweight object (chaff, sphere, starute etc.) floating through the atmosphere is a rather direct method to measure winds in the middle atmosphere. Still, there are several potential sources of uncertainties, such as the limited reaction time of the object to a wind changing with altitude, and restrictions due to the accuracy of the tracking radar. In this section we briefly revisit a simple method to estimate the time constant and the wind correction from the trajectory. We consider horizontal winds ($u$) only, i.e., we assume

that vertical winds are negligible compared to the fall velocity of the object. Ideally, the object follows the wind instantaneously and the winds are obtained from the trajectory $x(t)$ by $u = d/dt\{x(t)\} \equiv \dot{x}$ (for simplicity we assume that the object drifts in the x-direction only). In reality, however, the object reacts to a wind change with a certain time constant $\tau$ corresponding to an altitude resolution of $\Delta z_\tau$, which is sometimes called 'range constant' in the literature. Several theoretical studies have been performed to determine this time constant or, equivalently,

the wind correction required to account for the limited reaction time of the object to a wind changing with height (Hyson, 1968; Miller, 1969; Fichtl, 1971; Schmidlin, 1986). For this correction the drag coefficient of the object is needed which is a function of Reynolds (Re) and Mach (Ma) numbers ($Re = v \cdot \ell/\nu$, $v$ and $\ell$ are the velocity and size of the object, $\nu \sim 1/\rho$ is the kinematic viscosity). Under certain circumstances, drag coefficients can be measured in the laboratory.

The time constant introduced above can be estimated from the equations of motion:

$$
\ddot{z} = -g - \frac{C_z(z)}{m} \cdot (\dot{z} - w) = -g - \frac{1}{\tau_z(z)}(\dot{z} - w) \tag{1}
$$

$$
\ddot{x} = -\frac{C_x(z)}{m} \cdot (\dot{x} - u) = -\frac{1}{\tau_x(z)}(\dot{x} - u) \tag{2}
$$

where $F_z = C_z(z) \cdot \dot{z}$ and $F_x = C_x(x) \cdot \dot{x}$ are frictional forces in the z- and x-direction, and $\tau_z = m/C_z$ and $\tau_x = m/C_x$ are the time constants for the object following a change in wind. Furthermore, $g(z)$ is the acceleration

due to Earth's gravity, $w$ is the vertical wind, and $m$ is the mass of the object. Note that $C_x$ and $C_z$ are closely related to, but not identical to the standard definition of drag coefficients (see, e.g., Hyson, 1968). We estimate the time constant $\tau_x(z)$ from the measured deceleration in the vertical direction. We argue that $C_z(z)$ and $C_x(z)$ (and thereby $\tau_z$ and $\tau_x$) are rather similar at a given altitude. This is justified by noting that the drag coefficient does not vary much for Ma<1 and Re≫1 as can be seen (for a sphere) in Figure 2 in Lübken et al. (1994). The largest

variation in Re comes from the fact that the atmospheric density increases exponentially with height during the starute descent. With $\ell \sim 2$ m, typical velocities of 30-100 m/s, and kinematic viscosities of 0.01 and 0.001 m²/s at





50 and 30 km, respectively, Re is on the order of $10^3$ to $10^6$, i.e., significantly larger than unity. The relevant part of the starute flight therefore takes place in a regime where the drag coefficient varies little which means that we can assume that $C_x(z) \approx C_z(z)$ and $\tau_x(z) \approx \tau_z(z) \equiv \tau(z)$.

We derive the time constant $\tau(z)$ from the vertical equation of motion (eq. 1) under the assumption of zero vertical wind ($w = 0$), i.e., $\tau(z) = -\dot{z}/(\ddot{z} + g)$, and use this to calculate horizontal winds by $u = \dot{x} + \tau\ddot{x}$. The interpretation of $\tau$ is straight forward if we assume for a moment that $u = \text{const}$, $\tau = \text{const}$, and the starute is at rest at time zero. In this case we can integrate the equation given above and arrive at $\dot{x}(t) = u\left(1 - e^{-t/\tau}\right)$ which means that after a time $t = \tau$ the object has reached a speed of $(1 - 1/e) \cdot u = 0.63 \cdot u$ which is 63% of the actual wind speed. From the time scale $\tau$ we estimate a vertical reaction scale $\Delta z_\tau = -\frac{dz}{dt}\tau$ which is equivalent to the altitude resolution of wind measurements by this technique and which is called 'range constant' by other authors.

Figure 2 shows the time and range constants for a typical starute trajectory. To calculate the derivates a three-dimensional quintic spline was used. The analysis shows that starutes resolve vertical structures of horizontal winds on the order of 1 km at heights below 55 km and scales of less than 100 m below 40 km. At 60 km altitude the resolution degrades to ∼2 km and increases to about 12 km at 75 km. In Fig. 3 the results of the wind retrieval with and without the correction term are shown. As can be seen from this Figure the correction term amounts to less than 10 m/s around 55 km altitude and 5 m/s or less below that altitude. Immediately after starute ejection the corrections and corresponding uncertainties are particularly large due to the initial ejection velocity.

There is another potential error in determining winds from a starute, namely uncertainties introduced by the tracking radar (see, for example, Schmidlin and Michel, 1985). We have estimated this uncertainty from the standard deviation of starute positions obtained from the tracking radar in 100 m intervals. In our case this results in typical wind uncertainties of less than 0.2 m/s for most part of the trajectory, which can safely be neglected.

We note, that some starutes showed a somewhat abnormal behavior during flight and a rather abrupt variation in the radar return signal. We attribute this to a damaged or partial collapse of the starute and have considered wind measurements in this situation with caution because, for example, the fall speed may be higher than normal.

In summary, we arrive at uncertainties of winds from starutes of typically 5, 10, and >20 m/s at 50, 60, and 70 km, respectively. These values are similar to more sophisticated calculations published in the literature (see, for example, Hyson, 1968; Schmidlin, 1981, 1986). They are compatible with experimental studies of the response characteristics, repeatability, and the compatibility of different techniques (Miller and Schmidlin, 1971; Finger et al., 1975). Some decades ago, datasondes and so called 'falling spheres' (consisting of a one-meter diameter metalized sphere instead of a starute) were frequently used to characterize the background conditions during the launch of sophisticated instrumented sounding rockets (see, e.g., Schmidlin, 1985; Meyer et al., 1987; Lübken et al., 1990; Schmidlin and Schauer, 2001). These measurements were compared with various other techniques to measure winds, such as radar, chaff clouds, satellite borne instruments, and also from a Rayleigh Doppler lidar (Schmidlin, 1984; Gonzalez et al., 1994). In general, the wind uncertainties derived above are compatible with





these studies. Since the response deteriorates quickly above approximately 60 km we will not discuss in detail any potential differences between starute and DoRIS above this altitude.

## 4   Winds measured by DoRIS, datasondes, and radiosondes

In Fig. 4 we show the temporal development of the zonal and meridional wind field as measured by DoRIS. We
have also indicated the time/height lines for the datasonde and radiosonde flights. As can be seen from this Figure, the wind field consists of fluctuations down to rather small scales on top of a regular structure with, for example, quasi steady zonal wind maxima of up to +60 m/s at ∼30 km and ∼40 km altitude, and persistent but slowly descending meridional wind minima around 40–30 km, 60–50 km, and 70–65 km. The duration of the relevant part of the starute descend is typically 15-20 minutes, i. e., rather short in the time frame considered in Fig. 4.
On the other hand, radiosondes are in the air for up to 1–1.5 hours and travel typically 100–150 km kilometers horizontally before the balloon bursts.

In Fig. 5 we show all individual wind profiles measured by datasondes in that night. There are persistent wind maxima at 30 and 40 km (zonal winds) and at roughly 30-40 km (meridional winds) which are present from the first to the last flight, i.e., for a time period of at least 5-6 hours. These wind maxima are consistent with the
DoRIS wind field shown in Figure 4. Generally speaking both wind components tend to decline in magnitude with increasing height and are close to zero above approximately 60 km.

The relevant radiosonde flights are shown in Fig. 6. Only the uppermost part is relevant in the context of this paper. Despite the relative large time gap of 7.5 hours between the first and the last flight, the wind field is rather persistent. Because the deviations from one flight to another are rather small and are limited to small scales, and
because the distance of the volume sampled by the radiosondes relative to DoRIS and the datasondes is large, we decided to use a mean and smoothed radiosonde profile for further comparison (black line in Fig. 6). The zonal wind maximum at ∼30 km which was noticed in the DoRIS and datasonde profiles (see above) is also clearly visible in the radiosonde profiles.

The repeatability of wind profiles measured by DoRIS is demonstrated in Fig. 7 where all DoRIS profiles in
a period of ±30 minutes around datasonde flight SL6 are shown. More precisely we show the deviations of the DoRIS profiles from a profile heavily smoothed by spline fitting. As can be seen from Fig. 7 the wavy structure is very persistent in the one hour time period shown in this Figure. We argue that the major part of the fluctuations seen in Figure 7 is due to gravity waves. This is supported by the fact that the descent rates are roughly 10 km in 10 hours (see Figure 4) which is typical for the phase progression of gravity waves. In fact, we have shown
in an earlier paper that DoRIS observations are very suitable to study gravity waves (Baumgarten et al., 2016). The remaining deviations in Fig. 7 are rather small (typically 5-10 m/s) except for the uppermost heights above approximately 60 km. These small scale fluctuations are presumably due to a combination of small scale waves, turbulence, and instrumental noise.





In Fig. 8 and Fig. 9 we show two examples of a comparison between DoRIS, datasondes, and radiosondes. The datasonde profiles are shown together with all individual DoRIS profiles within a time period of ±20 minutes around the rocket flights. Furthermore, the mean radiosonde profile from Fig. 6 is also shown. As can be seen from Figures 8 and 9 there is excellent agreement between DoRIS and the datasondes, except for the lowermost part

(below approximately 30 km) where the presence of aerosols influences DoRIS winds (see next section). The very good agreement comprises not only the large scale zonal and meridional wind structure but also some larger scale modulations presumably caused by waves. We note that the experimental uncertainties of winds from datasondes are too large above ∼60-65 km to allow for a meaningful comparison with DoRIS.

## 5    Analysis of wind differences

In the following we analyze the difference between DoRIS and datasondes in more detail. In Fig. 10 we show the differences between datasonde profile SL6 and all individual DoRIS profiles shown in Fig. 8. The numbers in the plots give the mean of the differences and the root-mean-square deviations from the mean at certain altitudes. Typical mean deviations are 1–6 m/s, generally being smaller for the meridional (compared to the zonal) wind component. RMS values are on the order of 5–7 m/s and 3–9 m/s for the zonal and meridional wind deviations,

respectively, i. e., generally larger than the mean of the deviations. These numbers suggest that there is no systematic bias between DoRIS and datasondes, except for the lowermost altitudes (see below). The wavy structure of the deviations seen in Fig. 10 is presumably due to the fact that datasondes cannot resolve small scale structures above 40–45 km (e. g., caused by gravity waves) detectable by DoRIS and/or that different phase fronts of gravity waves are detected due to the horizontal distance of observations. Figure 10 confirms that mean deviations between

both techniques are generally small and are not systematic, i.e., there is no clear height structure in the deviations, except below ∼30  km (see later).

As can be seen in Figure 10, but also in Figures 8 and 9, the deviations between DoRIS and datasondes are smaller in the meridional wind component compared to zonal winds. This is presumably due to the fact that the datasondes are launched primarily in the northward direction. This means, that the northward directed lidar beam

is generally closer to the datasonde trajectory compared to the eastward directed lidar beam (see Fig. 1). The atmospheric volume used by DoRIS to measure meridional winds is therefore generally closer to the datasondes compared to the sampling for zonal wind detection.

Similar to Fig. 10 the differences for all datasonde flights are shown in Fig. 11. More explicitly, the differences between all datasonde wind profiles and the DoRIS wind profiles within a time period of ±20 minutes around each

particular datasonde launch are shown. We also show the overall mean profile of the differences as well as the root-mean-square variability of the differences relative to the mean difference profile. The mean difference is small for both wind components (typically less than ±5–10 m/s) and is consistently smaller than the RMS variability. This implies that, generally speaking, there is no systematic bias between DoRIS and datasondes, except for some





selected height ranges where the variability is rather large (e.g. at 35-40 km in the meridional wind component) and presumably due to natural fluctuations (waves etc.), i.e., due to different gravity wave phase fronts detected by DoRIS and by the starutes.

Fig. 11 highlights a systematic difference between DoRIS and radiosondes below roughly 30 km which was

earlier noted in the comparison between DoRIS and datasondes (see Figures 8, 9, and 10). This systematic difference is caused by the presence of aerosols. We have studied the backscatter ratio profiles in the visible ($\lambda$=532 nm) and in the infrared ($\lambda$=1064 nm) wavelength channel. The backscatter ratio $\beta$ is the relative difference between the received signal and the signal from molecular scattering only. A ratio larger than unity indicates the presence of aerosols. As can be seen from Fig. 12, backscatter ratios are significantly larger than unity for visible and

infrared wavelengths up to an altitude of approximately 30 km. This explains the systematic deviation between DoRIS and the radiosondes/datasondes below this altitude, as presented above. We note that $\beta$(1024 nm) is larger than $\beta$(532 nm) which is due to the fact that Rayleigh (molecular) scattering is smaller for larger wavelengths. In principle, it is possible to correct the wind measurements for the presence of aerosols. This requires, however, a sophisticated analysis of the contribution of the aerosol spectral line to the transmission characteristics of the

iodine line. Since aerosols only affect a small height range at the lower end of DoRIS profiles, we decided to leave this task for a later analysis.

## 6   Summary and conclusions

A new lidar technique called DoRIS was recently developed to measure winds (and temperatures) in the middle atmosphere. For a major part of the middle atmosphere this is the only technique to measure wind profiles nearly

continuously with high temporal and spatial resolution. The only alternative is to use rocket borne techniques, i.e., tracking the motion of an object falling through the atmosphere. This technique is well established since many years but can be applied sporadically only. In this paper we compare for the first time altitude profiles of zonal and meridional winds measured by DoRIS with observations deduced from 8 launches of so called 'starutes'. The measurements were made simultaneously and co-located at ALOMAR and from the Andoya Space Center at

69°N, respectively. Generally speaking there is excellent agreement between both techniques with typical mean differences of ±0–10 m/s. Most of the remaining deviations are of wavy nature and are most likely caused by the fact that DoRIS and the starutes detect different wave fronts of gravity waves. This comparison proofs that DoRIS is a reliable technique to measure winds in the middle atmosphere. This conclusion is highly non-trivial considering the complexity of the instrument required to address the challenge of measuring a relative Doppler

shift on the order of $10^{-7}$ to $10^{-8}$. DoRIS offers new capabilities for atmospheric physics since it allows for the first time to monitor mean winds (and temperatures) as well as gravity waves simultaneously with high temporal and spatial resolution in the entire middle atmosphere.





*Acknowledgements.* The support of the German MORABA team (DLR) and the Andøya Space Center for preparing and launching the meteorological rockets is highly appreciated. We thank Dr. Boris Strelnikov for leading the launch operation. This work was supported by the German Space Agency (DLR) under grant 50 OE 1001 (Project WADIS). DoRIS was partly supported by the European Union's Horizon 2020 research and innovation programme under grant agreement No 653980. The

5   authors wish to thank the National Aeronautics and Space Administration for providing the small meteorological rockets.



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



**Figures**

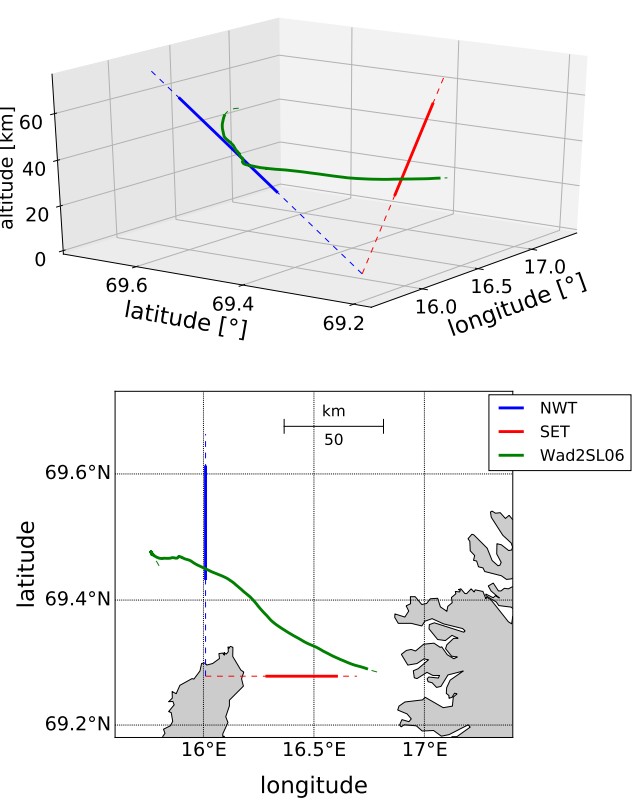

**Figure 1.** Trajectory of datasonde flight SL6 (green) and lidar beams for the NWT (blue) and SET (red) telescopes, respectively.
Thick lines represent the altitude range of 30 to 65 km.





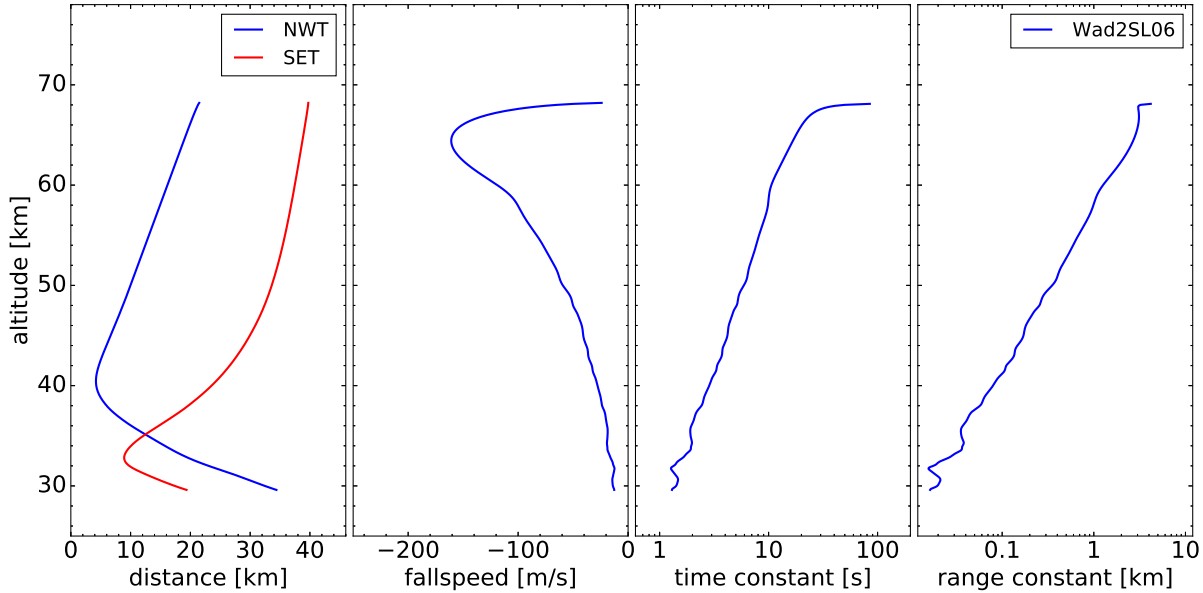

**Figure 2.** From left to right: Distance between the starute and the lidar beams for the two telescopes (NWT: blue, SET: red) ; fall speed of the starute ; time constant $\tau$ and range constant $\Delta z_\tau$ as defined in the text.

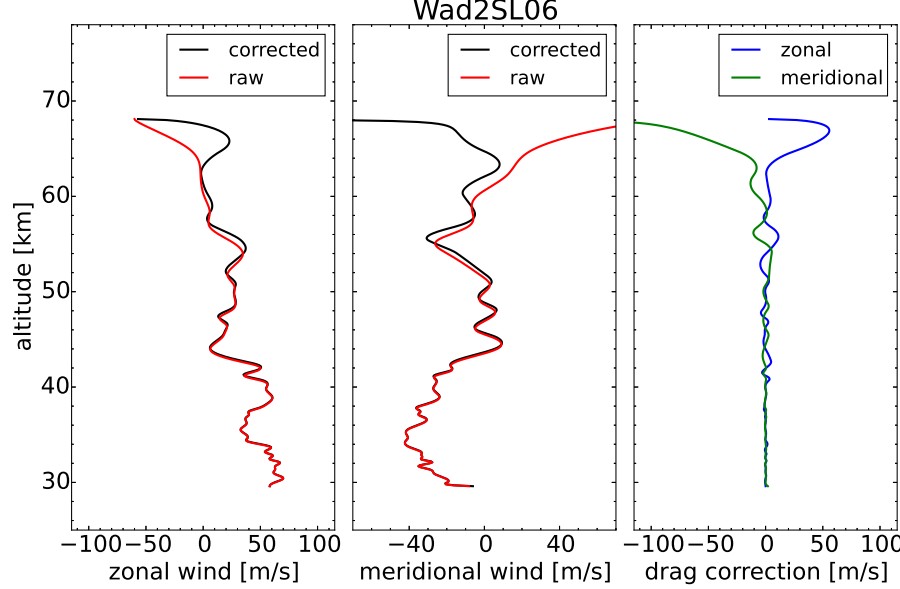

**Figure 3.** Zonal and meridional wind speed for SL6, without the wind correction (red) and including the wind correction (black) for the zonal (left panel) and meridional wind (middle panel) . The correction terms are plotted separately in the right panel. See text for more details.





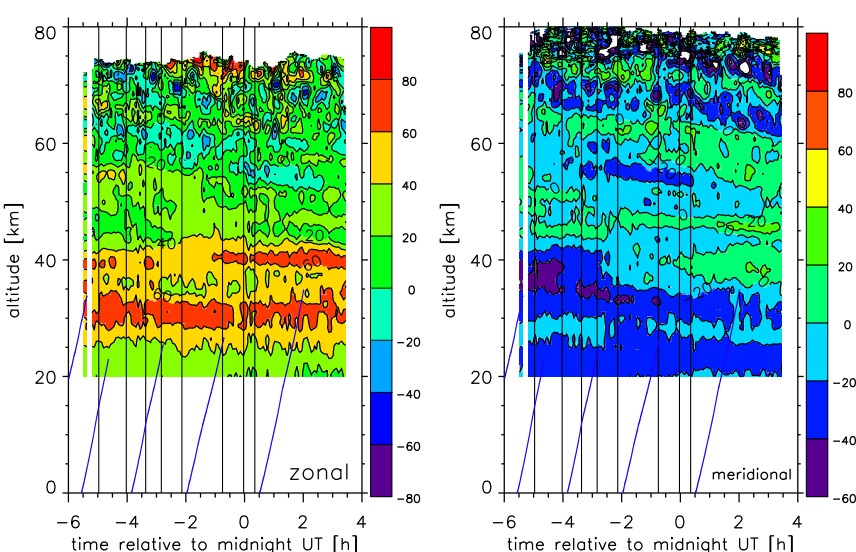

**Figure 4.** Contour plots of the wind field in m/s as measured by DoRIS on 14/15 March 2015. Left: zonal winds, right: meridional winds. The vertical lines mark the launch times of the datasondes. The slanted lines represent the time/height profiles of the radiosonde flights.





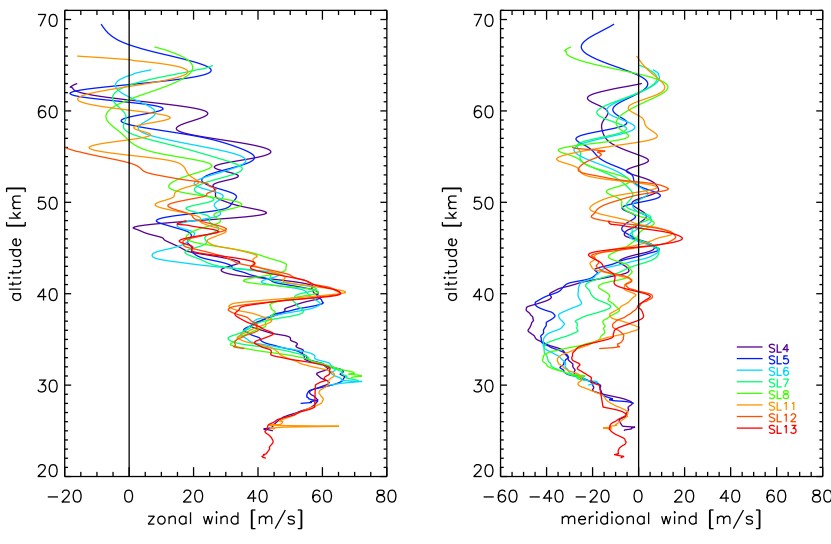

**Figure 5.** Individual wind profiles measured by datasondes on 14/15 March 2015. The inlet presents the color code of the individual flights. Left: zonal winds, right: meridional winds.





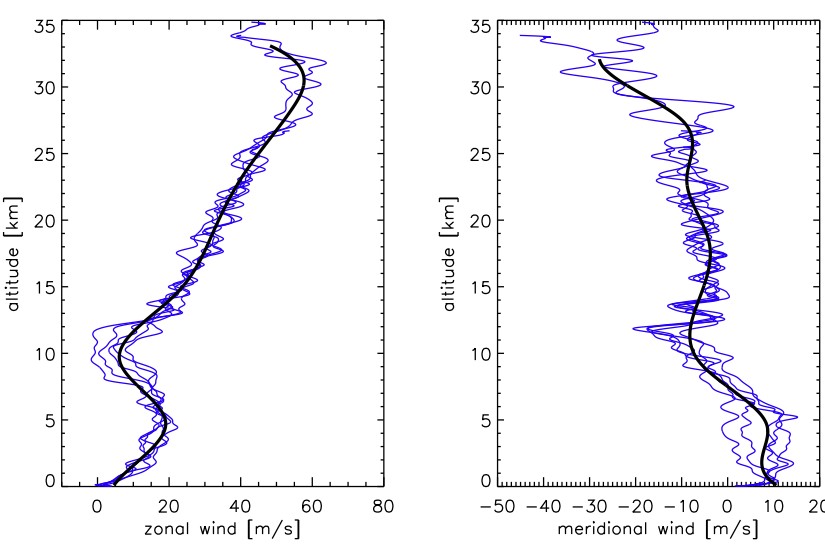

**Figure 6.** Radiosonde flights on 14/15 March 2015 relevant for the comparison with DoRIS. Blue: individual profiles, black: mean and smoothed profile. Left: zonal winds, right: meridional winds.





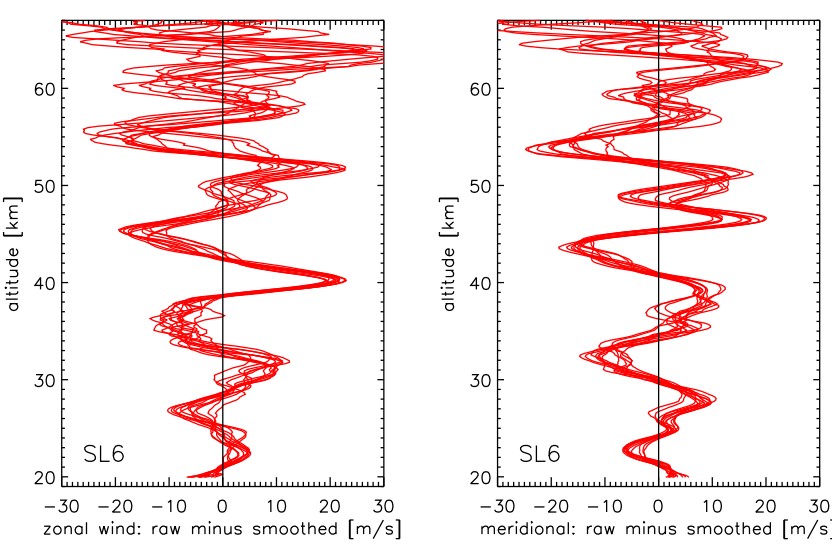

**Figure 7.** Repeatability of winds measured by DoRIS: All DoRIS profiles within a time period of ±30 min around datasonde flight SL6: difference between the individual profiles and a heavily smoothed profile. Left: zonal winds, right: meridional winds.



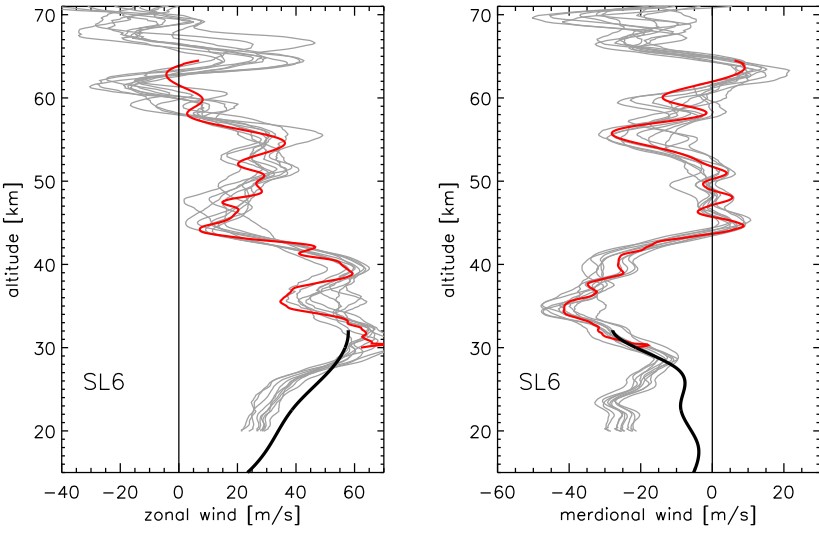

**Figure 8.** Wind profiles as measured by a single datasonde (SL6, red lines), the mean radiosonde profile (black line), and all DoRIS profiles within a period of $\pm 20$ min around the datasonde flight (gray lines). Left: zonal winds, right: meridional winds.





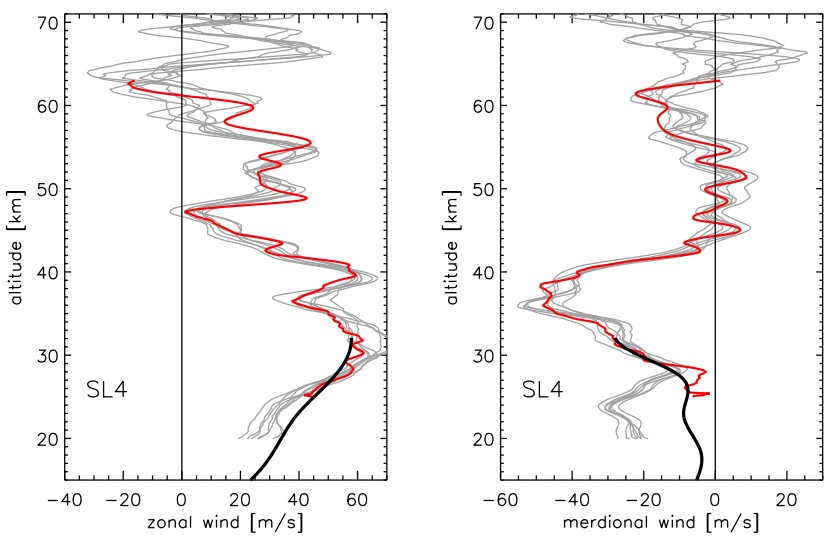

**Figure 9.** Same as Figure 8 but for datasonde flight SL4.





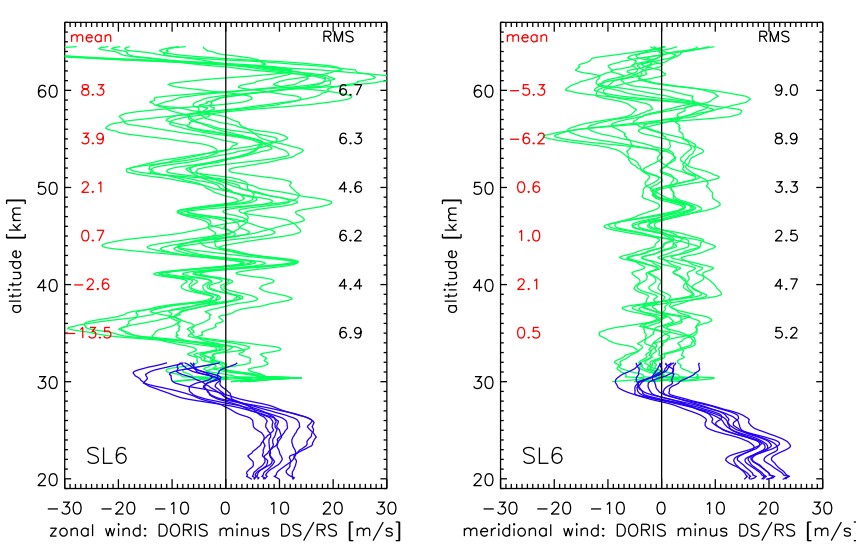

**Figure 10.** Differences between the DoRIS profiles shown in Figure 8 and i) the datasonde profile (green lines), and ii) the radiosonde profile (blue lines). See text for more details. Left: zonal winds, right: meridional winds.




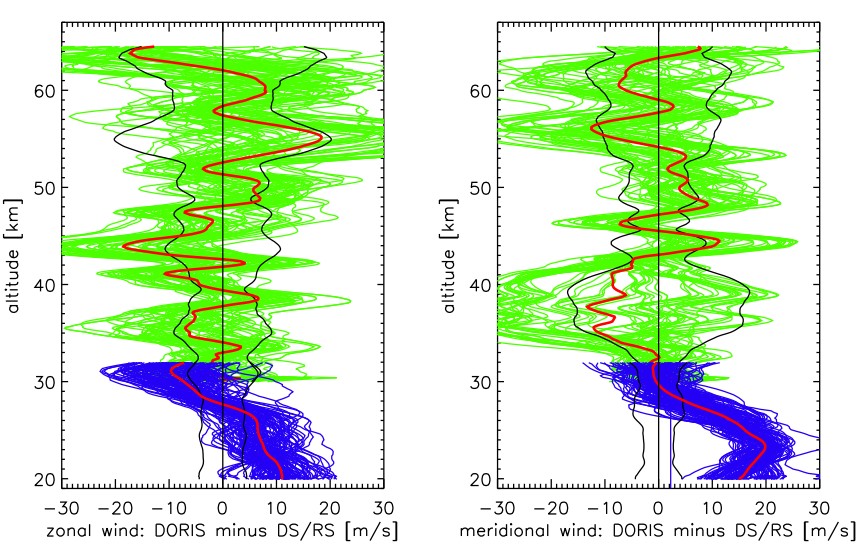

**Figure 11.** Green lines: differences between DoRIS profiles within a time period of ±20 min of a particular datasonde flight, shown for all datasonde flights. Blue lines: differences of all DoRIS profiles to the mean radiosonde profile shown in Figure 6. Red lines: mean of the differences. Black lines: root-mean-square variability of the differences relative to the mean difference. See text for more details. Left: zonal winds, right: meridional winds.





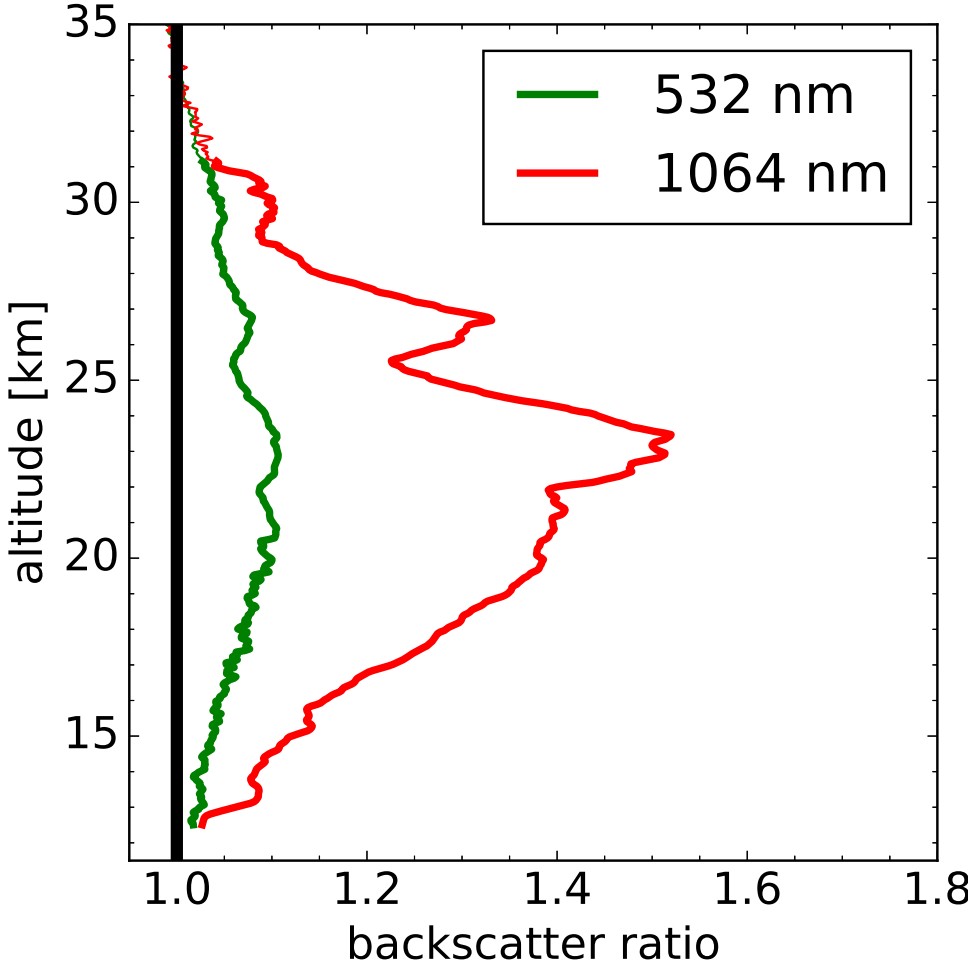

**Figure 12.** Altitude profiles of backscatter ratios $\beta$ for laser wavelengths of 532 nm (green line) and 1024 nm (red line). $\beta$ is the ratio of the total signal relative to the intensity expected from molecular scattering only. $\beta > 1$ indicates the presence of aerosols.



**Table 1.** List of datasonde launches during the WADIS-2 campaign.

| Mission-ID | Launch (UT) | Apogee (km) |
|---|---|---|
| **4/5 March 2015:** | | |
| WAD2SL01[a] | 22:16:00 | / |
| WAD2SL02[a] | 00:39:00 | / |
| WADIS2[b] | (01:44:00) | (126.05) |
| **10 March 2015:** | | |
| WAD2SL03 | 21:15:00 | 75.831 |
| **14/15 March 2015:** | | |
| WAD2SL04 | 19:00:00 | 77.20 |
| WAD2SL05 | 19:56:00 | 77.58 |
| WAD2SL06 | 20:36:00 | 68.24 |
| WAD2SL07 | 21:08:00 | 66.89 |
| WAD2SL08 | 21:50:00 | 70.55 |
| WAD2SL09[a] | 22:24:00 | / |
| WAD2SL10[a] | 22:47:00 | / |
| WAD2SL11 | 23:13:00 | 68.48 |
| WAD2SL12 | 23:56:00 | 78.02 |
| WAD2SL13 | 00:19:00 | 72.62 |

[a] = technical failure of rocket motor, starute ejection, or starute performance; [b] = the main instrumented sounding rocket.

**Table 2.** List of radiosonde launches.

| date | Launch (UT) |
|---|---|
| 14 Mar 2015 | 17:06:13 |
| 14 Mar 2015 | 18:26:27 |
| 14 Mar 2015 | 20:09:01 |
| 14 Mar 2015 | 22:02:10 |
| 15 Mar 2015 | 00:30:52 |