# Peer review of "Simultaneous and co-located wind measurements in the middle atmosphere by lidar and rocket-borne techniques"

_Atmospheric Measurement Techniques, 2016_

## Referee Comment (RC1) · Anonymous Referee #1 · 29 Apr 2016

**Review of manuscript «Simultaneous and co-located wind measurements in the middle atmosphere by lidar and rocket-borne techniques» by Lübken, Baumgarten, Hildebrand, and Schmidlin.**

General comments: This excellent and well-written manuscript compares measurements results of horizontal winds in the height region 20 to 75 km. For the first time, lidar measurements enhanced with a technique called Doppler Rayleigh Iodine Spectrometer (DoRIS) are compared with rocket-borne "starute" measurements, as well as with datasondes and radiosondes. The techniques are described, including their advantages and disadvantages. The most important conclusion is that DoRIS is a reliable technique to measure winds in the middle atmosphere, with the caveat that care must be taken to exclude data from heights where aerosol scattering is present. On the day of the comparison that was below about 30 km.

Figures 1 to 4 make it easy for the reader to understand what is being compared. All figures are of high quality, including symbol sizes and the choice of line colors.

Specific comments:

(1) I recommend a paragraph discussing the **Eulerian** nature of the lidar measurements (wind measurements at a fixed geometric position in the atmosphere), the **Lagrangean** nature of the radiosondes (the weather balloon drifts with the atmospheric flow and measures the wind at that time and place), and the intermediate nature of the starutes and datasondes. Given these differences of reference frame, the very good agreement is almost surprising. In a wind field that is constant in space and time, Eulerian and Lagrangean measurements ought to give exactly equal results. If the wind field changes in space and/or in time, even ideally precise measurements in Eulerian and in Lagrangean reference frames come out differently. Is it possible to estimate how much of the (small) remaining differences shown in Figures 8 to 11 might be due to the different reference frames? If it is, we can learn even more about the atmospheric flow characteristics.

(2) As this manuscript and other publications show, measurements and analysis techniques have vastly improved in recent decades. I submit that it is time we also moved on from the **assumption of zero vertical wind.** Although there likely were no measurements of the vertical wind w during the observations for this manuscript, we do know in a general sense what average w to expect on time scales of many hours and spatial scales of 100 km and more: mm/s to cm/s (e.g., Körner and Sonnemann, 2001; Berger and von Zahn, 2002). We also know, in a general sense, what w fluctuations to expect locally and on time scales of tens of minutes: typically $\pm 3$ to $\pm 5$ m/s but not more than $\pm 10$ m/s (e.g., Fritts et al., 1990; Hansen and Hoppe, 1996; Collis, 1997). Most of the references I have found here pertain to the summer mesosphere and many to altitudes higher than those required for this manuscript. Nevertheless it must be possible to estimate the influence of such typical vertical winds on the observations in this manuscript. – At zenith angle α, an actual vertical wind w would appear as a line-of-sight wind $w_{apparent} = w \cdot \cos \alpha$ . When this line-of-sight wind is misinterpreted as a horizontal wind, it would appear as

$u_{apparent} = w_{apparent} \cdot \sin \alpha = w \cdot \cos \alpha \cdot \sin \alpha = \dfrac{w}{2} \cdot \sin 2\alpha$ . Assuming $w = 3 \dfrac{m}{s}$ and $\alpha = 30°$ ,

we obtain $u_{apparent} = 1.5\frac{m}{s} \cdot \sin 60° \approx 1.3\frac{m}{s}$. With $w = 3\frac{m}{s}$ and $\alpha = 20°$, we obtain

$u_{apparent} = 1.5\frac{m}{s} \cdot \sin 40° \approx 1\frac{m}{s}$. Therefore, features in the meridional wind in Figures 7 to 11 up to 1.3 m/s may be attributed to a non-zero vertical wind, and up to 1 m/s in the zonal wind in those figures. This argumentation is for the vertical wind variance observed in other publications on time scales of tens of minutes. It is easy to see that the much smaller average vertical wind (mm/s to cm/s) gives effects smaller than the graphic linewidth in those figures. – A similar argumentation ought to be carried out concerning page 5 line 5, the zero-vertical-wind assumption in the starute analysis.

(3) It might seem worthwhile to plot Figure 7 as a hodograph in addition. Such a hodograph would presumably show whether the variations are due to one or several gravity waves, and also their vertical propagation direction. It is possible that the authors have tried this, and that it turned out not to be worth showing.

(4) On page 8, lines 6-12 in section 5, the authors discuss the backscatter ratio. I did not find the source of the signal from molecular scattering only. This is presumably from the $N_2$ Raman channel of the lidar. It should be mentioned.

Technical corrections:

(1) The authors may wish to consider excluding the datasonde launches of 4/5 March and 10 March from Table 1, as they are not used in this manuscript. There may be a good reason to keep them in the table, even if this reviewer does not know it.

(2) Typos:
    a. P. 2 l. 8: satellite-borne
    b. P. 2 l. 31: atmosphere
    c. P. 3 l. 24: lowercase "figures"
    d. P. 5 l. 29: so-called
    e. P. 6, l.9: descent
    f. P. 6., l. 15: consider "decrease" instead of "decline"
    g. P. 8 l. 20: rocket-borne
    h. P. 8 l. 27: proves

(3) Word explanation:
    a. P. 3, l. 20: burble fence: perhaps a figure or a bit more detailed explanation of this unusual word?
    b. P. 6, l. 6: "on top of" can be understood in two ways; consider "on"

(4) Commas before subordinate clause (new grammatical subject) starting with "which":
    a. P. 1 l. 16
    b. P. 2 l. 9
    c. P. 2 l. 19
    d. P. 2 l. 23
    e. P. 3 l. 33
    f. P. 4 l. 17
    g. P. 5 l. 2

(5) Commas that may be removed:

       a. P. 5 l. 22, after "We note"

       b. P. 7 l. 24, after "This means"

(6) Other correction:

       a. P. 8 l. 21: consider "This technique has been well established for many years…"

**References:**

Berger, U. and U. von Zahn, Icy particles in the summer mesopause region: Three-dimensional modeling of their environment and two-dimensional modeling of their transport, JGR, 107, A11, 1366, doi: 10.1029/2001JA000316, 2002.

Collis, P., An improved technique to determine neutral winds in the auroral mesosphere using the EISCAT VHF incoherent scatter radar, JASTP, 59, 15, 1909-1918, 1997.

Fritts, D.C., U.-P. Hoppe and B. Inhester, A study of the vertical motion field near the high-latitude summer mesopause during MAC/SINE, JATP, 52, 10-11, 927-938, 1990.

Hansen, G. and U.-P. Hoppe, Investigation of the upper mesospheric dynamics under late polar summer conditions by EISCAT and lidar, JATP, 58, 1-4, 317-335, 1996.

Körner, U. and G.R. Sonnemann, Global three-dimensional modeling of the water vapor concentration of the mesosphere-mesopause region and implications with respect to the noctilucent cloud region, JGR, 106, D9, 9639-9651, 2001.

---

## Referee Comment (RC2) · Anonymous Referee #2 · 20 Jun 2016

General comment:

As the title is saying the paper compares simultaneous and co-located wind measurements in the middle atmosphere by lidar and rocket borne techniques. Data have been obtained at the arctic location of Andoya at 69° north during one night in March 2015. The altitude range covered where both techniques deliver reliable data extends from approximately 30km to 65km. In case of the lidar horizontal wind is obtained by analyzing the Doppler-shifted backscattered signal at a wavelength of 532nm whereas in case of the rocket sondes the backscattered radar signal of a starute is analyzed. In addition also wind information beyond approx. 30 km altitude is obtained from a conventional radio sounding.

[Figure]

Wind measurements in the middle atmosphere are very sparse and every new technique or comparisons of techniques are very valuable.

The paper is clearly written with adequate reference to the literature. All the figures are of high quality and illustrate clearly what has been deduced from the data analysis.

The paper merits publication with some minor corrections.

Specific comments:

The comparison of horizontal wind profiles is based on data from one night only. Though it is understandable that rockets are only launched during specific circumstances it would be of interest to know how the techniques would compare during day. How well is the lidar technique suited to retrieve the wind profile during day. Also it would be interesting to know how well the technique would compare when the edge of the vortex is close to the observation site. The reviewer would appreciate if the authors could spend a few sentences about these aspects even if they do not have any data to compare with.

The authors give quite some details about the uncertainty analysis in case of the floating radar target in the atmosphere (drag coefficients etc). They also indicate an uncertainty due to radar tracking and refer to a publication of a conference that the reviewer was unable to find. Two or three sentences here explaining what is the problem would be helpful.

The lidar is observing with quite a small zenith angle (20 and 30 degrees) for the detection of horizontal wind. Is there any possibility to retrieve vertical wind? Is there any possibility to have the lidar beams with higher zenith angles in order to reduce errors?

Also the reviewer who is not familiar with rocket sondes would have been grateful to find a small description of the concept of a starute or even see an image of these devices. The authors give reference to papers but most of them are very old and not accessible

through the web. It took the reviewer some time to finde a description of what a starute is.

Page 5, line 25: Please also indicate the uncertainty of the starute measurements at 30 and 40km altitude.

The only paragraph where the reviewer was not able to follow the argumentation of the authors is on page 6 and Figure 7 dealing with the repeatability of the profiles measured by the lidar. Deviations of the individual wind profiles from a heavily smoothed one by spline fitting is shown. I just do not see what the authors want to tell here. What kind of profile is the smoothed one.

Minor comments:

The first sentence in the abstract says that a comparison of the lidar technique with data from insitu observations is performed. Either techniques are compared or data from different techniques but not a technique and data. Please rephrase.

page 2 second line: the covered altitude range of the lidar and the rocket is closer to 30km and 65 km. If the authors also consider the radio soundings then the altitude range is of course larger. However they state upper stratosphere. So 20km and upper stratosphere somehow do not match.

---

## Author Comment (AC1) · 12 Jul 2016

**Reviewer No. 1**

Response to the report of the reviewer to our paper:

'Simultaneous and co-located wind measurements in the middle atmosphere by lidar and rocket-borne techniques´

by Franz-Josef Lübken, Gerd Baumgarten, Jens Hildebrand, and Francis J. Schmidlin

submitted for publication in *Atmos. Meas. Tech.*, 2016.

Manuscript Number: amt-2016-106

**Introductory remarks:**

We appreciate the comments from the reviewer. We have taken his/her suggestions for improvements into account when preparing the revised version of the manuscript. In the following we respond to the reviewer's comments point by point. We have marked the changes in the revised version of the manuscript.

1. We fully agree with the reviewer who has summarized correctly the fundamental difference between lidar (Eulerian) and insitu (Lagrangian) observations. Following his/her suggestions we have inserted a comment on this topic at the beginning of section 5.

2. We thank the reviewer for addressing this topic in such detail. He/she is correct in stating that there are no vertical wind observations available during our campaign. Even worse, we don't know of any technique measuring(!) vertical winds in the mesosphere/upper stratosphere. Some few exceptions basically involve radar techniques (mainly EISCAT) in the upper mesosphere/lower thermosphere and the foil cloud technique introduced by Hans Widdel ('Vertical movements in the middle atmosphere derived from foil cloud experiments', *J. Atmos. Terr. Phys.*, 49, 723–742, 1987). Both techniques suffer from significant uncertainties. The other examples cited by the reviewer are from models and may be not be correct or not be applicable in our situation. On a long term average (more than some hours) vertical winds are presumably very small (mm/s to cm/s) and are mainly determined by the residual circulation. As the reviewer pointed out correctly, we can safely ignore these small vertical winds. On shorter terms, vertical winds are expected to be significantly larger, for example as generated by gravity waves. Therefore, some of the gravity wave features detected in our observations may indeed be due to vertical winds. As mentioned in the paper, we will perform a more detailed analysis of the gravity wave signatures in a later paper. For starutes and radiosondes, we will have to consider gravity wave modulations of background densities also, since they modify vertical movements (we will apply

the polarization relations for gravity waves for this purpose). In summary, the suggestion by this reviewer is well taken and will be considered in a future paper. We have added a note on this topic in the revised version of our paper.

3. It is indeed a good idea to characterize the apparent gravity wave features in more detail using hodograph methods. Again, we refer to the main aim of our paper, namely a comparison of instrumental techniques, whereas a detailed analysis of the entire wave field will be performed in a later paper. We hope for the understanding of the reviewer. We have added a short note regarding the hodograph technique.

4. The reviewer is correct: scattering due to air molecules only (without aerosols) was derived from Raman scattering on $N_2$ at 608 nm. We have added a note in the manuscript.

Technical corrections

(1) The reviewer is correct that the launches from 4/5 March and 10 March are not used in this paper. Still, we would like to keep them in Table 1 because i) it demonstrates the overall success rate of the starute flights, and b) the Table may be used for future reference.

(2) We have corrected the typos.

(3) a.: In the revised version we have added a drawing of a starute where the 'burble fence' can easily be identified. The purpose of this device is explained in the main text.
b.: We have changed the wording.

(4) We have inserted commas, as suggested.

(5) We have removed commas, as suggested.

(6) We agree and have changed the wording.

Kühlungsborn, July 11, 2016.                    Franz-Josef Lübken (for all coauthors)

Response to the report of the reviewer to our paper:

'Simultaneous and co-located wind measurements in the middle atmosphere by lidar and rocket-borne techniques´

by Franz-Josef Lübken, Gerd Baumgarten, Jens Hildebrand, and Francis J. Schmidlin

submitted for publication in *Atmos. Meas. Tech.*, 2016.

Manuscript Number: amt-2016-106

**Introductory remarks:**

We appreciate the comments from the reviewer. We have taken his/her suggestions for improvements into account when preparing the revised version of the manuscript. In the following we respond to the reviewer's comments point by point. We have marked the changes in the revised version of the manuscript.

*Specific comments:*

- *'Comparison during daylight'*
  We have performed a few flights during daylight conditions some years ago but haven't put too much emphasis on analyzing these flights (yet) due to some technical problems with the starutes and also with DoRIS at that time. We think, however, that winds from DoRIS are also reliable during daylight conditions, since there is no apparent 'jump' in the wind field at the transition between day and night, and vice versa (see Figures 1 and 3 in the Baumgarten et al. paper from 2016). We have added a short note on this topic.

- *'Uncertainty due to radar tracking'*
  The basic information on the position of the starute comes a radar tracking the starute. If that radar track is uncertain, e. g. because of improper identifying and following the target, this results in errors in positioning the starute and thereby in uncertainties of the winds derived from the trajectory. A deeper analysis is given in the Schmidlin et al. paper mentioned in the paper (this manuscript is part of the ESA proceedings from a long-standing symposium on rocket and balloon borne techniques). In our case, we have estimated this uncertainty to be on the order of ±0.5–1.5 m/s, depending on the performance of the starute.

- *'Vertical winds from DoRIS?'*
  DoRIS can in principle measure vertical winds if we point one (or better both) telescopes into the vertical direction. However, mean vertical winds are much

smaller compared to horizontal winds and vertical winds cannot be obtained from starutes (only horiontal winds). We therefore pointed the telescopes as much as possible to the horizontal. The maximum zenith angle of the telescopes is 30 degrees and is given by mechanical constraints of the supporting structures.

- *'Datasondes'*
  We have added a drawing of a starute in the revised version which hopefully makes it easier to understand how this system works. The basic performance and the purpose of the 'burble fence' is explained in the main text.

- *'Uncertainty at 30/40 km'*

  As requested, we have added a note regarding the error bars below 40 km

- *'What is shown in Figure 7'*
  We wanted to demonstrate the natural variability of the wind field as measured by DoRIS within a certain time period of $\pm 30$ min around a given datasonde flight (SL6 in the case of Figure 7). We think that this is an important information when considering any potential difference between DoRIS and starute winds since these measurements are not made at exactly the same location and not exactly at the same point in time (e.g., due to time averaging of the DoRIS profiles). We have replaced the word 'repeatability' by 'natural variability' in the text.

*Minor comments*

- *'First sentence in the abstract'*
  We agree with the reviewer and have modified the first sentence in the abstract.

- *'upper stratosphere ?'*
  We agree and have added 'middle' in the revised version.

Kühlungsborn, July 11, 2016.                    Franz-Josef Lübken (for all coauthors)